# A mutation-induced drug resistance database (MdrDB)

Ziyi Yang[1,2], Zhaofeng Ye[1,2], Jiezhong Qiu[1], Rongjun Feng [1], Danyu Li[1], Changyu Hsieh[1], Jonathan Allcock[1] & Shengyu Zhang [1✉]

Mutation-induced drug resistance is a significant challenge to the clinical treatment of many diseases, as structural changes in proteins can diminish drug efficacy. Understanding how mutations affect protein-ligand binding affinities is crucial for developing new drugs and therapies. However, the lack of a large-scale and high-quality database has hindered the research progresses in this area. To address this issue, we have developed MdrDB, a database that integrates data from seven publicly available datasets, which is the largest database of its kind. By integrating information on drug sensitivity and cell line mutations from Genomics of Drug Sensitivity in Cancer and DepMap, MdrDB has substantially expanded the existing drug resistance data. MdrDB is comprised of 100,537 samples of 240 proteins (which encompass 5119 total PDB structures), 2503 mutations, and 440 drugs. Each sample brings together 3D structures of wild type and mutant protein-ligand complexes, binding affinity changes upon mutation ($\Delta\Delta G$), and biochemical features. Experimental results with MdrDB demonstrate its effectiveness in significantly enhancing the performance of commonly used machine learning models when predicting $\Delta\Delta G$ in three standard benchmarking scenarios. In conclusion, MdrDB is a comprehensive database that can advance the understanding of mutation-induced drug resistance, and accelerate the discovery of novel chemicals.

[1] Tencent Quantum Laboratory, Shenzhen 518057 Guangdong, China. [2]These authors contributed equally: Ziyi Yang, Zhaofeng Ye. ✉email: shengyzhang@tencent.com

Structural mutation of proteins can directly affect their folding and stability[1–3], function[4,5], interactions with other proteins[6,7] and binding affinity[8]. In some cases, it can result in significant perturbations or even complete abolishment of protein function, potentially leading to diseases or cancer[9,10]. The evolutionary pressure imposed by small molecule drugs on many quickly evolving systems, including cancer cells, viruses, and bacteria, can lead to the rapid development of resistance[11–14]. While novel and cheap high-throughput sequencing technologies have made it possible to identify mutations in large populations[15,16], the significance and characteristics of any novel polymorphisms currently require time-consuming and expensive experiments to determine[17]. Protein-ligand binding affinity data are of great value for understanding the impact of polymorphisms on disease and identifying mutations that lead to drug resistance[18,19]. Convenient and broad access to such data for wild-type and mutant proteins would aid our understanding of the mechanisms of mutation-induced drug resistance, increase the accuracy of extrapolations to novel mutations and systems, enable more effective computational approaches for drug resistance prediction, and facilitate the development of combination therapies and the discovery of novel chemicals.

Fortunately, a number of databases on the effects of mutation on protein-ligand binding affinity have been compiled and released in recent years. In particular, Platinum[17] was the first database to provide experimental data on changes in protein-ligand affinities upon mutation, along with three-dimensional structures, while tyrosine kinase inhibitor (TKI)[18] contains reliable inhibitor ΔpIC50 data for 144 clinically identified mutants of the human kinase Abl. Together, these two datasets are the current gold standards, both providing well-curated data and three-dimensional structural information. Other resources include Auto In Silico Macromolecular Mutation Scanning (AIMMS)[20], RET[21] and kinase mutations and drug response (KinaseMD)[22], which provide kinase inhibitor resistance information. However, while the value of these datasets is significant, there remain a number of deficiencies. Platinum and TKI are restricted to known cocrystal structures and thus contain relatively few samples (approximately 1000 and 150, respectively), which can limit their use with machine learning models. For AIMMS, RET, and KinaseMD, structural information is not provided. In addition, while all of the aforementioned databases consider single-point and multi-site substitution mutations, they do not include more varied and complex mutation types—such as deletions, insertions, and insertion-deletions (indel)—which play an important role in certain disease progressions.

To tackle these challenges, we have created MdrDB, an all-encompassing database that significantly expands the amount of information on drug resistance. MdrDB combines the five databases mentioned above and achieves considerable augments by incorporating large-scale drug response data, and somatic mutation information from various cancer cell lines, as reported in the Genomics of Drug Sensitivity in Cancer (GDSC) dataset[23], and Cancer Dependency Map (DepMap)[24]. Further supplemented with data from supporting databases such as RCSB Protein Data Bank (PDB), UniProtKB and PubChem, and 3D structures computed with PyMOL and AlphaFold 2.0[25], MdrDB brings together 3D structures of wild-type and mutant protein-ligand complexes, binding affinity changes upon mutation (ΔΔG), and biochemical features calculated from complex structures. MdrDB contains 100,537 samples, generated from 240 proteins (5119 total PDB structures), 2503 mutations, and 440 drugs. A total of 95,971 samples are based on available PDB structures, and 4566 samples are based on AlphaFold 2.0 predicted structures.

MdrDB offers several key advantages over existing publicly available protein mutation databases. Firstly, MdrDB offers massive and diverse data. With over 100,000 samples, MdrDB integrates information from multiple sources, includes various mutation types beyond single site-point mutation and covers mutations across a broad range of protein families, which is the largest mutation-induced drug resistance database to our knowledge. Secondly, MdrDB provides 3D structural information on all wild-type and mutant proteins along with 146 associated biochemical features, which are useful and important in accurate drug resistance modeling. Furthermore, we evaluated the classical drug resistance prediction machine learning models' performance on three standard benchmarking scenarios. By using MdrDB as a training set, nearly all models gain significant performance improvement. In summary, MdrDB has the size, breadth, and complexity to offer new resources for protein mutation studies and drug resistance modeling.

## Results

**Database statistics**. At the time of writing, MdrDB contains 100,537 samples, generated from 240 proteins (5119 total PDB structures), 2503 mutations, and 440 drugs. In all, 95,971 of the total samples are based on available PDB structures, with the remaining 4566 samples predicted with AlphaFold 2.0. In addition to single-point substitution mutations and multiple-point substitution mutations, MdrDB also contains complex mutations including deletion, insertion, and indel (insertion-deletion) mutations, as well as multiple-site mutations containing a number of the aforementioned mutations. Table 1 summarizes the data contained in MdrDB, and an overview of the data sources is shown in Supplementary Table 1. Figure 1A shows MdrDB samples categorized by their mutation types. Approximately 83.6% of the samples are single substitution mutations; 11.9% are multiple substitution mutations; and 4.5% are complex mutations, of which deletion mutations account for the largest proportion.

The distribution of mutation-induced ligand binding affinity changes, measured by ΔΔG, is shown in Fig. 1B. A standard for defining resistant samples was given in[18], where samples with a larger than 10-fold drop in binding affinity, corresponding to ΔΔG >1.36 kcal mol$^{-1}$, are considered to be resistant. Based on this standard, 8197 (8.2%) samples in MdrDB are mutation resistant.

We analyzed the distribution of amino acid types in substitution mutation samples, before and after mutation, to shed light on mutation preferences. The 20 natural amino acids were divided into five groups: positively charged, negatively charged, polar, hydrophobic, and three amino acids classified as special cases (Fig. 1C, Supplementary Fig. 1A, and Supplementary Tables 2 and 3). Overall, 30.8% of mutations are observed within the same amino acid group, while 69.2% of mutations occur across different amino acid groups. Arginine (R) is the most frequently mutated amino acid, followed by glycine (G) and valine (V), which are reported to contribute to human genetic diseases[26]. Leucine (L), alanine (A), serine (S) and cysteine (C) are the most frequently mutated amino acids. Similar conclusions hold for the MdrDB_CoreSet (Supplementary Figs. 1B and 2 and Supplementary Tables 4 and 5), where non-repetitive samples considering only unique "protein-drug-mutations" are kept.

The mutation frequencies of different amino acids at the mutation sites are highly correlated with the frequencies calculated based on codon frequencies (Pearson $r = 0.945$, Supplementary Figs. 3A–C and Supplementary Table 6)[26]. However, the frequencies of amino acids after the mutation show a lower correlation with the frequencies calculated based on codon frequencies (Pearson $r = 0.633$, Supplementary Figs. 3D–F)[26]. Many complex factors could contribute to this, such as epigenetic

**Table 1 Overview of data represented in MdrDB.**

| Property | No. of counts | No. of counts in source datasets |
|---|---|---|
| No. of samples | 100,537 | AIMMS (5048), DepMap (1500), GDSC (91,215), KinaseMD (1334), Platinum (840), RET (456), TKI (144) |
| No. of unique UniProt IDs | 240 | AIMMS (10), DepMap (5), GDSC (99), KinaseMD (31), Platinum (126), RET (1), TKI (4) |
| No. of unique ligands | 440 | AIMMS (16), DepMap (5), GDSC (180), KinaseMD (48), Platinum (126), RET (3), TKI (8) |
| No. of mutations | 2503 | AIMMS (121), DepMap (49), GDSC (1805), KinaseMD (530), Platinum (43), RET (11), TKI (31) |
| No. of unique UniProt ID-Mutations | 2673 | AIMMS (173), DepMap (49), GDSC (1818), KinaseMD (43), Platinum (578), RET (11), TKI (80) |
| No. of unique PDB IDs | 5119 | AIMMS (430), DepMap (220), GDSC (4128), KinaseMD (802), Platinum (241), RET (21), TKI (8) |
| Mutant structure source | | No. of Platinum processed 840; No. of TKI processed 144; No. of PyMOL mutated 94,987; No. of AlphaFold 2 folded 4566 |
| Drug pose source | | No. of Platinum cocrystal 840; No. of TKI cocrystal 131; No. of TKI docked 13; No. of docked 99,553 |
| Mutation types | | No. of single substitution 84,038; No. of multiple substitutions 11,977; No. of deletion 3228; No. of insertion 355; No. of indel 243; No. of complex 696 |

modifications[27], disease preferences[28], co-evolution[29], and experimental assay biases[30]. In addition, we analyzed the mutation spectrums for each type of amino acid, which revealed mutation preferences. Interestingly, as shown in Supplementary Fig. 4, most of the amino acids have a mutation spectrum that is similar to the one based on codon frequencies (Pearson $r > 0.8$). For the amino acids which deviate the most from this expected spectrum, a higher ratio proportion of lysine (K), glutamine (Q), cysteine (C), and tryptophan (W) are mutated to A, which may come from the loss of function mutation experiments, such as alanine scanning mutagenesis[30]. We carried out the same analyses on susceptible/resistant samples (Supplementary Fig. 5), and observed an obvious decrease in the frequency correlation of resistant samples before mutation (Pearson $r = 0.874$), where serine (S), lysine (K), and tyrosine (Y) showed the largest differences. For the frequency of each amino acid after mutation in the susceptible /resistant samples, although the correlations are similar, their distributions vary greatly.

Furthermore, we analyzed the secondary structure elements where mutations occurred across all samples (Supplementary Fig. 6) as well as in resistant samples (Supplementary Fig. 7). As shown in Supplementary Figs. 6 and 7, the $\Delta\Delta G$ value distributions were not significantly different across samples whether mutations are located in $\alpha$-helices, $\beta$-sheets or loops.

Annotations for the proteins and drugs are provided for each sample in MdrDB. For proteins, a total of 160 kinds of protein domains, 154 protein families and 143 protein superfamilies can be found in MdrDB. The protein domains with at least 100 samples are shown in Fig. 1D and Supplementary Table 7. Protein kinase domains account for the largest proportion (39.5%) of all samples, followed by DNA-binding domains (18.5%), retroviral matrix proteins (4.8%) and bromodomains (4.7%). For drug mechanism annotations, a total of 20 FDA-documented pharmacological classes, 60 Medical Subject Headings (MeSH) pharmacological classes and 9 PubChem drug classes can be found in MdrDB. 67 (15.2%) out of all 440 drugs have known FDA pharmacological classes, and the corresponding samples were counted and shown in Fig. 1E and Supplementary Table 8. The largest proportion of samples takes kinase inhibitors as drugs (12.0%), followed by HIV-1 reverse transcriptase inhibitors (4.5%).

**Model performance evaluation.** Rapid and accurate computational methods could impact clinical decision making by providing oncologists with an initial indication of whether an observed protein mutation may lead to drug resistance to certain inhibitors. Molecular dynamics (MD)-based free-energy calculations and physics- and knowledge-based potential scoring functions (e.g., Rosetta) are two common types of computational methods for estimating affinity changes in proteins with mutations[31–36]. Although these two types of methods can achieve remarkable performance and serve as a gold standard, they suffer from high computational overheads. Data-driven machine learning methods have recently been developed to predict the impact of ligand binding affinity changes upon protein mutations and identify resistance-causing mutations[19,37], and can make predictions within seconds once the features associated with protein mutation affinity changes are input. Current machine learning methods, however, are prone to overfitting, and limited training data—expensive to acquire—reduces their performance stability in real applications. For instance, Aldeghi et al.[19] employed extremely randomized regression trees to predict TKI affinity change values. Their model—trained on a subset of Platinum database (no tyrosine kinase) with 484 training samples and then tested on the TKI dataset—achieved low performance in

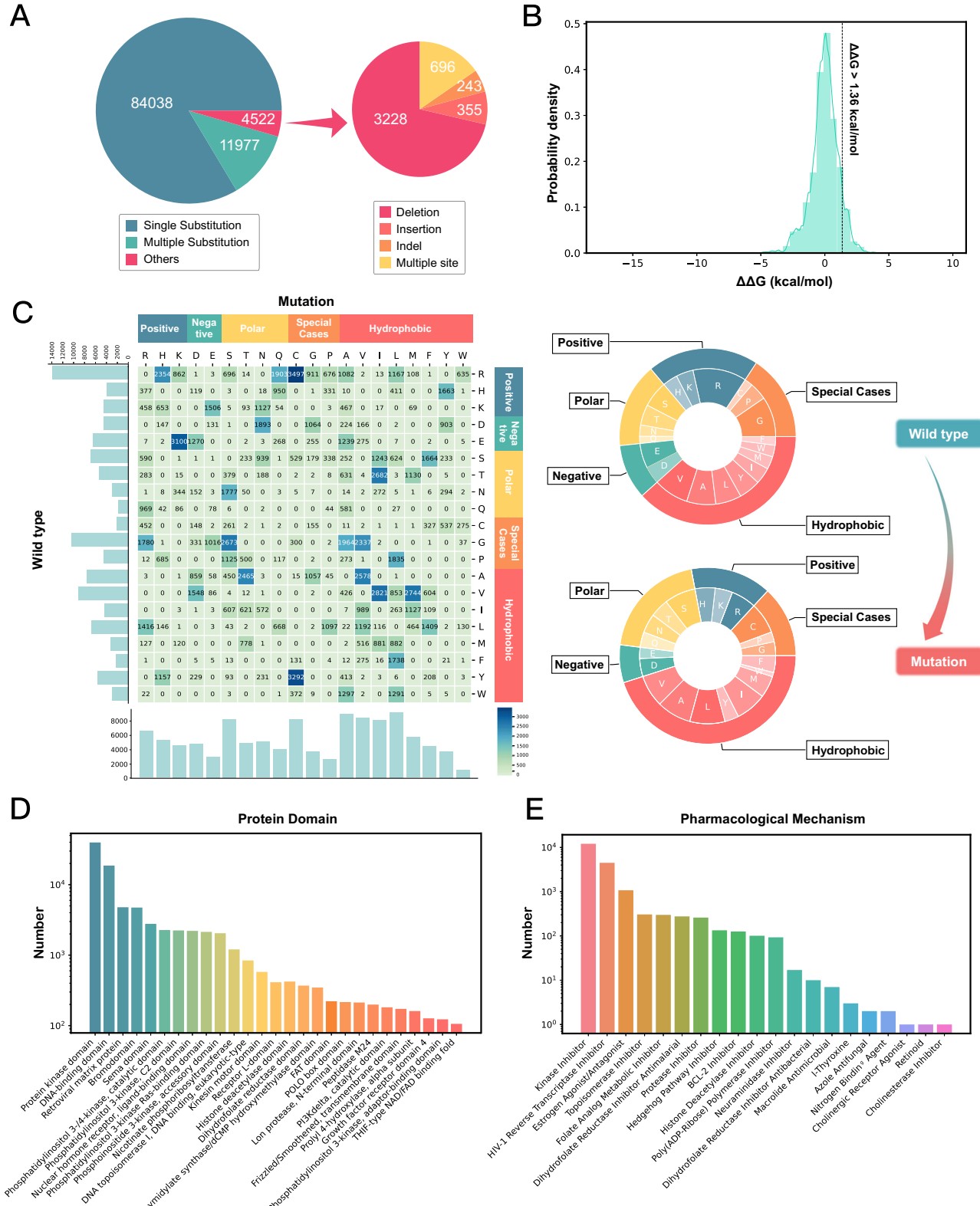

terms of correlation and classification ability. To alleviate the overfitting problem, our previous work[38] incorporated additional physics-based structural features and protein family information as inputs to learn more comprehensive knowledge from limited training data. The empirical results show that the method we previously proposed is slightly superior to MD simulations with the Amber force fields. In this section, we investigated three

scenarios to test whether using the MdrDB_CoreSet as the training dataset (using 146 calculated biochemical features) can improve the model generalization performance in predicting mutation-induced drug resistance in the TKI dataset.

We conducted experiments on the TKI dataset under three scenarios. The first scenario was to train the machine learning methods on the subset of the Platinum dataset, which had no

**Fig. 1 MdrDB mutation statistics, ΔΔG distribution, and protein and drug annotations. A** Number of samples of each mutation type, where a sample corresponds to a (Uniprot, PDB, mutation, drug) combination. "Others" includes four mutation types: deletion, insertion, indel, and complex. **B** Histogram of protein mutation-induced ligand binding affinity changes measured by ΔΔG (kcal mol$^{-1}$). Line at ΔΔG = 1.36 kcal mol$^{-1}$ separates mutations defined as resistant from susceptible. **C** Number of amino acid changes from substitution mutations. Left: heatmap shows the number of amino acid changes from substitution mutations, with different colors corresponding to the number of changes in different amino acids. The number of samples for each amino acid in the wild-type (mutation) is displayed as a bar chart along the left axis (bottom) of the plot. Right: donut charts show the number of amino acids in the samples corresponding to substitution mutations. The proportion of each amino acid in wild-type (mutated) samples is displayed top (bottom). **D** Number of samples annotated by protein domains. X-axis gives protein domain, with Y-axis the corresponding sample number on logarithmic scale. **E** Number of samples annotated by pharmacological mechanisms.

tyrosine kinase samples, and then test them on the TKI dataset. This setting is consistent with the setting in work[19,38]. The second scenario was to train the machine learning methods on the Platinum dataset, and test them on the TKI dataset. The third scenario was to train the machine learning methods on a subset of MdrDB_CoreSet, removing samples derived from the TKI dataset and selecting samples corresponding to single substitutions, and then test them on the TKI dataset. Since the TKI dataset contains only the single substitution type, we select samples belonging to a single substitution in MdrDB_Coreset. Compared with Scenario 1, the latter two scenarios evaluate whether adding a small amount of tyrosine kinase information or increasing samples and protein family information in the training dataset can improve the capability of the model in predicting affinity changes in Abl kinase mutants. Detailed information on the training datasets for the three scenarios is given in Supplementary Table 9. According to the feature calculation section, 146 features were calculated as model inputs, carrying potentially useful information for predicting affinity changes introduced by protein mutations.

Four families of methods were used to evaluate the drug resistance prediction performance under three scenarios. The first family is tree-based methods, including decision tree (DecisionTree)[39], random forest (RandomForest)[40], and extremely randomized regression trees (ExtraTrees)[41]. The second family is linear-based methods containing support vector regression (SVR)[42], elastic net linear regression (Elastic Net)[43], and lasso regression (Lasso)[44]. The third family of baselines is ensemble-based methods including bagging regressor (Bagging)[45], AdaBoost[46], and gradient boosting (GradienBoost)[47]. The fourth family is neural network-based methods such as multi-layer perceptron[48].

Root mean square error (RMSE), Pearson correlation coefficient (Pears), and the area under the precision-recall curve (AUPRC) were used to evaluate the model performance under three training scenarios[19]. Consistent with the previous work[19,38], resistant mutations are defined as the affinity changes for mutants by least 10-fold, i.e., $\Delta\Delta G_{exp}$ >1.36 kcal mol$^{-1}$.

We train the machine learning methods on different training datasets (i.e., Platinum (no tyrosine kinase), Platinum, and MdrDB_Coreset (single substitution)) and report the test performance averaged over five repetitions for each method on the TKI dataset in Fig. 2A. We can clearly see that the machine learning methods obtained poor performance in terms of correlation and classification ability when training on Platinum (no tyrosine kinase). For instance, ExtraTrees achieved low accuracy on TKI dataset with weak correlation (Pearson = 0.075), and poor classification performance (AUPRC = 0.198). When the training dataset provided a small amount of tyrosine kinase information, i.e., training on the Platinum dataset, the test prediction performance of most machine learning methods improved slightly. ExtraTrees obtained slightly better estimates (RMSE = 0.907, Pearson = 0.094, and AUPRC = 0.243) compared with its training on Platinum (no tyrosine kinase). When the machine learning methods were trained on the subset of MdrDB_CoreSet, we found that the prediction accuracy of most

machine learning methods was improved on the TKI dataset compared to the previous two scenarios, except for AdaBoost, DecisionTree, and SVR. In particular, ExtraTrees, GradienBoost, Bagging, and RandomForest have significantly improved prediction performance. For instance, ExtraTrees achieved highly accurate on the TKI dataset when training on the subset of MdrDB_CoreSet with low absolute errors (RMSE = 0.656 kcal mol$^{-1}$), strong correlation (Pearson = 0.607), and good classification performance (AUPRC = 0.538). This result outperformed MD simulations with the Amber force fields (e.g., A99 and A99l) reported in the work[19] by a considerable margin, demonstrating that MdrDB_CoreSet could improve the generalization of the machine learning method in predicting affinity changes in Abl kinase mutants. Figure 2B shows the scatter plots of the experimental versus calculated ΔΔG values of each machine learning method obtained at a certain time by training on the subset of MdrDB_CoreSet and testing on the TKI dataset. The scatter plots of the experimental versus calculated ΔΔG values in the first and second scenarios are shown in Supplementary Figs. 8 and 9.

In addition, we also provide a comprehensive evaluation of 10 common machine learning models in several different scenarios (Supplementary Figs. 10–23 and Tables 10–23), and provide baseline prediction results on the MdrDB database. It is our hope that this will help further the development of new machine learning algorithms using the MdrDB database and facilitate drug resistance research. Please refer to Supplementary Note 1 for details.

## Conclusion

Here we have introduced MdrDB, the largest drug resistance database to provide all information highly relevant to protein mutation-induced drug binding affinity changes. In addition to basic information regarding the proteins, drugs, mutations and changed affinity, and structural data for wild-type and mutant complexes, 146 calculated biochemical features and extra annotations are also provided. These features and annotations were chosen to be convenient to use for model training and sample splitting in machine learning algorithms for drug resistance prediction[38]. In addition, MdrDB is also the first database that includes drug resistance-related mutation types beyond substitution mutations, and the availability of wider and more complex mutation types can be used to test the generalization of machine learning models.

A comprehensive resource providing a variety of information for studying protein mutation, predicting drug resistance and discovering novel chemical compounds, MdrDB will be updated regularly and include more public data in the future.

## Methods

The MdrDB database construction pipeline is shown in Fig. 3. Data preparation includes four steps: (1) data collection from the seven publicly available datasets. (2) Data preprocessing and integration of information from other databases, so that each sample contains information on the protein, drug, mutation and ΔΔG. (3) 3D structure generation for the wild-type and mutant protein-drug complexes. (4) Calculation of biochemical features based on the complex structures.

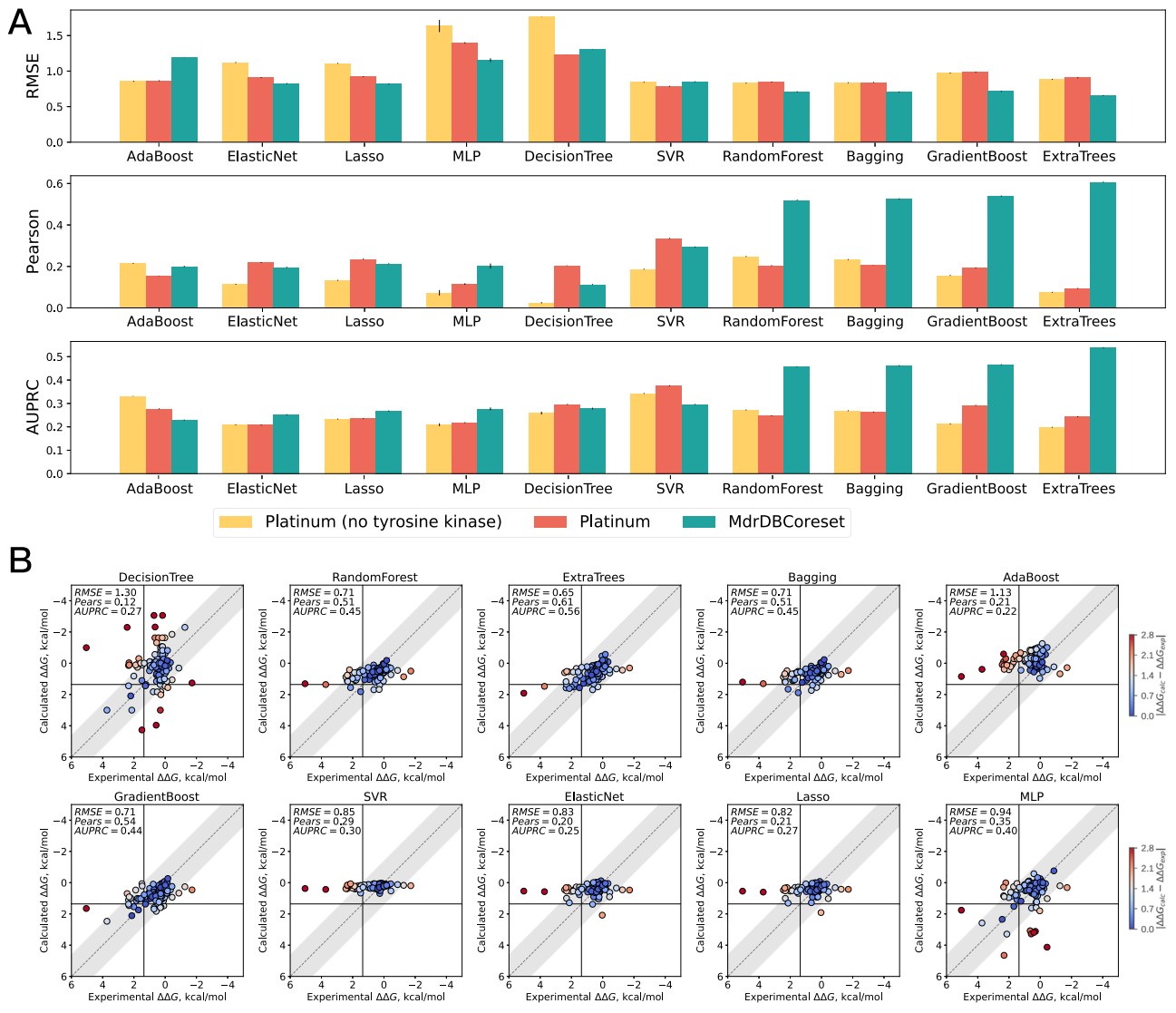

**Fig. 2 Machine learning methods performance evaluation. A** Summary of the test performance of the $\Delta\Delta G$ prediction across machine learning methods under three training scenarios in terms of RMSE, Pearson correlation, and AUPRC. Means with error bars (standard deviation) for all competing methods. **B** Scatter plots of the experimental versus calculated $\Delta\Delta G$ values in Scenario 3. $X$-axis denotes the experimental $\Delta\Delta G$ values (kcal mol$^{-1}$). $Y$-axis denotes the calculated $\Delta\Delta G$ value (kcal mol$^{-1}$). Each $\Delta\Delta G$ estimate is color-coded according to its absolute error *w. r. t.* the experimental $\Delta\Delta G$ value; at 300 K, the 1.4 kcal mol$^{-1}$ error corresponds to a 10-fold error in the $K_d$ change and 2.8 kcal mol$^{-1}$ error corresponds to a 100-fold error in the $K_d$ change.

**Data collection**. MdrDB contains data from seven publicly available datasets: Platinum[17], AIMMS[20], TKI[18], RET[21], KinaseMD[22], GDSC[23], and DepMap[24]. These datasets differ slightly in terms of the information they contain.

- Platinum: provides data on affinity changes upon site mutation ($\Delta\Delta G$) and experimental cocrystal structures of protein-drug complexes from the RCSB PDB. It contains more than 1000 manually curated mutations[17].
- AIMMS: a web server for predicting site mutation-induced drug resistance, and contains a dataset of 17 protein-ligand complexes involving 311 $\Delta\Delta G$ values and mutations[20].
- TKI: a database of TKIs resistances in ABL tyrosine kinase site mutations. A total of 144 $\Delta\Delta G$ values are included, along with ABL-TKI complex structures from PDB[18].
- RET is a dataset specifically related to drug resistances between three drugs (cabozantinib, lenvatinib, vandetanib) and the RET kinase mutants. 56 IC50 measurements and protein-drug complex structures were reported in the research[21].
- KinaseMD: focuses on kinase mutations, and integrates information from the Cancer Cell Line Encyclopedia (CCLE)[49] and GDSC[23] databases, and provides various annotations of drug responses on kinase mutants. In all, 79 $\Delta\Delta G$ values were collected from this database[22].
- GDSC: the largest public resource for information on drug sensitivity in cancer cells. The database contains 4.4 million drug sensitivity (IC50)

values across 518 drugs and 1000 cancer cell lines. It also records information related to basal expression, mutation, copy number variation, and gene methylation in the cell lines.

- DepMap: similar to GDSC, and contains drug sensitivity data on cell lines from the Achilles[50] and CCLE projects[51].

For Platinum, AIMMS, TKI, RET and KinaseMD, mutation information and mutation-induced binding affinity changes are directly obtainable from the datasets. However, for GDSC and DepMap, extra steps are required to obtain the mutations and $\Delta\Delta G$ values (see next section).

**GDSC/DepMap raw data processing**. Mutation and $\Delta\Delta G$ values were obtained from these two datasets via a process similar to that used by KinaseMD: (i) mutation information acquisition; (ii) drug-cell line response calculation; and (iii) $\Delta\Delta G$ calculation.

Cell lines were first grouped into wild-type and mutated samples for a specific protein, and mutation information for the protein was gathered. Specifically, for a particular protein, cell lines that did not have mutations on them were considered to be control (wild-type samples), while cell lines with mutations were considered to be mutant. Mutations were then filtered to retain only six types of mutation in terms of amino acid changes in protein sequences:

- Single substitution mutation: replacement of one amino acid in a protein with a different amino acid.

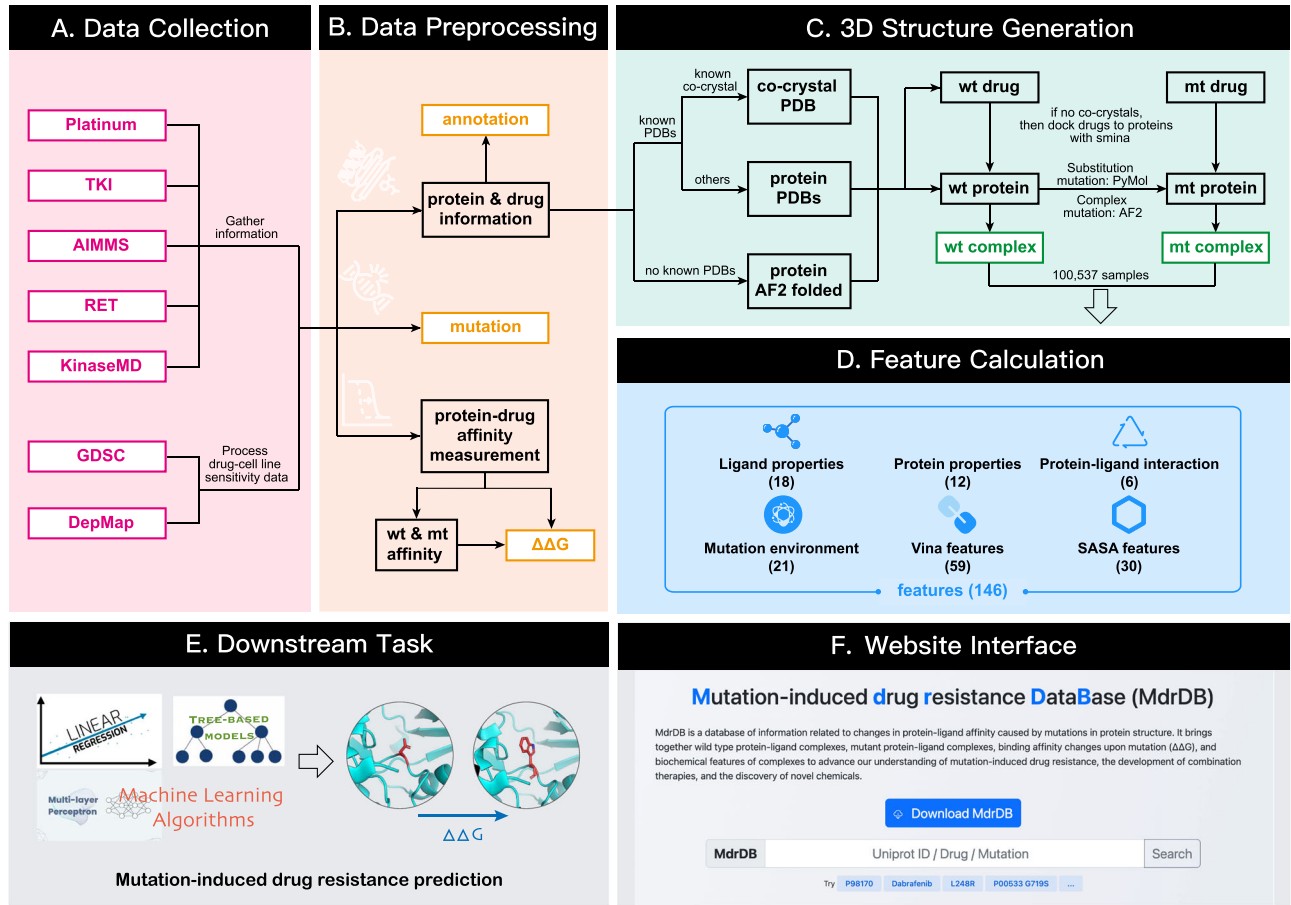

**Fig. 3 The MdrDB database construction pipeline. A** Data collection from public datasets. **B** Data preprocessing to extract sample information. **C** 3D structure generation of protein-drug complexes for both wild-type and mutant proteins. **D** Feature calculation. **E** Downstream task: mutation-induced drug resistance prediction. **F** Website construction for browsing, search and downloading of the data. Color-coded boxes: red—the collected publicly available datasets, the number of original samples is shown; yellow—annotations, mutation information and $\Delta\Delta G$ values, provided in MdrDB in .tsv file format; green—structural information provided in MdrDB.

- Multiple substitution mutation: more than one amino acid substitution in a protein.
- Deletion mutation: removal of one amino acid or an amino acid sequence in a protein.
- Insertion mutation: addition of one amino acid or an amino acid sequence in a protein.
- Indel mutation: a deletion mutation followed by an insertion mutation, i.e., replacement of an amino acid sequence in a protein with another amino acid sequence.
- Complex mutation: combinations of the above five amino acid mutation types.

Then, we filtered the drugs with known protein targets and queried drug sensitivity (IC50 values) data on all cell lines. For drugs with multiple known protein targets, each protein was considered individually. If, in the cell line, only one of these proteins was mutated, the cell line was kept. Otherwise, the cell line was discarded. After this, for each (protein, cell line, drug) mutant sample, the mutation string was generated by merging all mutations of the mutant protein to obtain a (protein, cell line, drug, mutation) sample. Then IC50 values were averaged over all cell lines for the (protein, cell line, drug, mutation) sample following the sensitivity data processing in KinaseMD[22]. This average value was taken as the mutant protein-drug affinity value IC50(mut). A corresponding wild-type sample was similarly assigned and IC50s again averaged over cell lines, and this average value was taken as the wild-type protein-drug affinity IC50(wt).

Finally, these affinity pairs were used to calculate $\Delta\Delta G$ according to the formula[18,37]:

$$\Delta\Delta G_{\mathrm{exp}} = RT\ln\frac{K_{i,\mathrm{mut}}}{K_{i,\mathrm{wt}}} \approx RT\ln\frac{\mathrm{IC}_{50,\mathrm{mut}}}{\mathrm{IC}_{50,\mathrm{wt}}} \quad (1)$$

**Sample data consolidation**. From all seven datasets, five basic pieces of information were collected and consolidated: protein name, UniProt ID, mutation string, drug name, and $\Delta\Delta G$ value. UniProt IDs were identified by querying UniProtKB[52] with protein names. The consolidated data was then divided into separate tables, with samples of the same mutation type kept together. For samples from Platinum and TKI, wild-type and mutant protein-drug complex structures were also obtained from the original datasets. For protein annotations, we used the Interpro API[53] to query the protein family, homologous superfamily and domain information. For drug annotations, we used the PubChem PUG REST API[54] to query the CID, Depositor-Supplied Synonyms, FDA mechanism, MeSH and Drug Classes information.

**Three-dimensional structure generation**. With the exception of samples from Platinum and TKI, which already directly include 3D structure information, we prepared 3D structures of wild-type proteins, mutant proteins and drug binding poses for all samples from the other datasets.

For the preparation of wild-type protein structure, first for each protein (UniProt ID), all associated PDBs were identified with the RCSB REST API[55]. The .pdb or .cif for 3D structures and .fasta for sequences were downloaded from RCSB PDB. Then, all water, solvent, and ions were removed from the PDB files. Next, the protein chains and ligands were split into separate files. Each chain was annotated and only the chains corresponding to the UniProt ID were kept. If multiple chains were found to exist for protein, the longest one was kept. Meanwhile, the largest ligand was kept as the docking box generation reference. Finally, each mutation for a protein was checked against all available PDBs. If the mutation sites could be found in the PDB, the mutation and drug would be assigned to the PDB.

For the mutant protein structure generation, we generated mutant structures from wild-type structures using PyMOL[56] or from wild-type sequences using AlphaFold 2.0[25]. Specifically, the Mutagenesis Wizard module of pymol-open-source v2.5.0 was used to generate a mutation by replacing a residue with a new amino acid type, sample several rotamers from the rotamer library and generate several non-clashing conformations. Then, the most likely rotamer was chosen as the mutated residue. For AlphaFold 2.0, we used the protein amino acid sequence

as the input to predict the structures. A length threshold of 2000 was set for computing resource considerations. For Multiple Sequence Alignment searching, "reduced_db" was used. For inference, the "monomer_ptm" models were used. Five models were generated and the one with the highest averaged pLDDT (predicted lDDT-C$\alpha$) value was chosen as the predicted structure for further procedures. Several rules were followed to decide which tool was used for mutant structure generation. For proteins with known PDBs containing the mutation sites:

- For single substitution and multiple substitution mutations, PyMOL was used for mutant protein generation.
- For deletion, insertion, indel and complex mutations, AlphaFold 2.0 was used for mutant protein structure prediction. For fair comparison and feature calculation, post-processing was carried out to keep the residue numbers the same except at the mutated sites.

For proteins with no known PDBs containing the mutation sites, AlphaFold 2.0 was used for both wild-type protein and mutant protein structure prediction. Structures with an average pLDDT larger than 70 for the whole structure were kept, which was a confidence threshold for the predicted structures in AlphaFold 2.0. In addition, if a mutated site was located in a region that was poorly predicted (pLDDT < 50), the sample was also discarded. After structure generation, the mutant protein was aligned with the wild-type protein. This alignment was carried out by only taking the backbone atoms of both wild-type and mutant residues whose pLDDTs were larger than 70 into consideration.

For drug structure generation, SMILES strings for all drugs were first obtained using the PubChem PUG REST API[54] (several drugs that could not be directly identified via PubChem were manually checked and assigned). Ions and salts were then removed from the SMILES, the structures neutralized, and the resulting SMILES strings rewritten in canonical format. 3D structures were first generated using Open Babel 3.1.1[57], using the "–gen3D" flag, and non-polar hydrogens were added to the generated structures with the "–addpolarH" flag. After structure generation, molecular docking was carried out with smina[58], using default docking parameters:

- If the wild-type PDB contained a known in-pocket ligand, then "–autobox_ligand" was selected.
- If no ligand was present, the whole protein was used to generate the docking box.

After docking, the conformation with the best smina score was kept.

**Feature calculations**. Based on the structures of the wild-type and mutant protein-ligand complexes, we calculated a total of 146 biochemical features relevant to machine learning prediction of mutation-induced affinity changes. These features were first used in the work of Aldeghi et al.[19], and we follow the procedures in their original paper for their calculation:

- Eighteen ligand properties such as logP, molecular weight, number of hydrogen acceptors and donors were calculated using RDKit.
- Twelve features describing the changes for the mutated amino acids were calculated (again with RDKit), such as hydrophobicity, number of heavy atoms, and the change in side-chain volume.
- Twenty-one features describing the mutation environment were calculated with Biopython, including the distribution of ligand and protein atoms around the mutation site and the number of residues in different property groups.
- Six features related to the protein-ligand interactions were calculated using the Protein-Ligand Interaction Profiler[59]: hydrogen bonds, hydrophobic contacts, salt bridges, $\pi$-stacking, cation-$\pi$ interactions, and halogen bonds.
- The Vina binding score as well as 58 scoring function terms were calculated using Delta-Vina XGBoost[60,61].
- Thirty pharmacophore-based solvent accessible surface area features are calculated using Delta-Vina XGBoost[61].

With the exception of the ligand property features, the numerical values $x$ of all other features given in MdrDB correspond to the difference between the mutant ($x_{mt}$) and wild-type values ($x_{wt}$), i.e., $x = x_{mt} - x_{wt}$.

**Website interface**. The web interface of MdrDB was implemented using caddy2 and Bootstrap v5.0. All processed data, structure files, biochemical features and annotations were stored in Tencent Cloud Object Storage. Interactive charts were implemented using Apache ECharts. A full tutorial for MdrDB is available at https://quantum.tencent.com/mdrdb/tutorial. A user-friendly website provides access to the curated data and structure information on wild-type and mutant complexes, and provides functions for browsing, searching, displaying and downloading the data. For detailed information on the constructed website, please see Supplementary Note 2: web design and interface. Supplementary

Fig. 24 shows the search page, browse page and sample display page of the MdrDB website.

## Data availability
All data are available to browse and download on the MdrDB website: https://quantum.tencent.com/mdrdb/.

## Code availability
The code is available at https://github.com/tencent-quantum-lab/MdrDB/.

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

## Acknowledgements

We are grateful to our colleagues in the Tencent Quantum Laboratory for helpful feedback and suggestions on the construction of the MdrDB website.

## Author contributions

Z.Y.Y. and Z.F.Y. collected, preprocessed, and analyzed all the data. J.Z.Q. used Alpha-Fold 2.0 to predict the structures of proteins. Z.Y.Y. calculated a total of 146 biochemical features and evaluated the performance of the current machine learning methods. R.J.F., D.Y.L., Z.Y.Y., and Z.F.Y. designed and developed the MdrDB website. Z.Y.Y., Z.F.Y., and J.A. wrote and reviewed the manuscript. C.Y.H., and S.Y.Z. read the manuscript. All authors reviewed the manuscript.

## Competing interests

The authors declare no competing interests.
