## [Peer Review File · Communications Chemistry]

A mutation-induced drug resistance database (MdrDB)Reviewers' comments:

Reviewer #1 (Remarks to the Author):

Yang et al. present a new database (MdrDB) containing protein-ligand binding affinity changes for a set of ~2500 mutations across >200 proteins. MdrDB was created by consolidating data from 7 datasets with a custom data processing pipeline. Structural information was included using experimental data from the Protein Data Bank or AlphaFold2 predictions. Pre-computed features that can be used as input for machine learning models are also provided, which is useful to create a standardized benchmark for descriptor-based models. The significance of this effort lies in the creation of a consistent and machine-readable dataset of mutational affinity values that is considerably larger than existing datasets, and one that includes multiple and complex (e.g., insertions and deletions) mutations. Overall, the manuscript is clear, concise, and well written.

For the above reasons, my opinion is that this work is in principle publishable in Communications Chemistry. At the same time, I felt that the analysis of the dataset and, in particular, the exploration of ML models was fairly superficial and with room for improvement. My two main suggestions are the following:

(1) While MdrDB contains over 200 protein targets and ~2500 mutations, only a small subset (the TKI set with ~140 mutations) was used to evaluate ML models. While the TKI work is interesting to show how more data results in a better model, this seems like a missed opportunity to utilize the newly created dataset to evaluate the ability of ML to predict drug resistance more broadly across protein classes and mutation types (single, multiple, complex).

Many additional interesting insights may arise from such a benchmark, e.g., is there a difference in predictive performance between mutations with experimental vs predicted structures? Which kind of mutations are more challenging to predict with ML? How far from the training set are these models able to extrapolate? One can imagine splitting the dataset in a few different ways to test ability to extrapolate in protein, ligand, or mutation space. These are only a few examples of possible questions one could explore with MdrDB. In addition, creating multiple train/test splits (like in cross validation) would allow to compute uncertainties for the performance metrics used (i.e., adding error bars to the plots in Figure 3A).

I understand that a comprehensive study may be beyond the scope of this work, but I think it would still be worth exploring a couple of hypotheses while making use of the full dataset.

(2) It would be useful to expand the analysis of the dataset to discuss the statistics of mutations for the class of resistant mutations only and how that differs from the distribution of non-resistant mutations. For example, at page 7 it was noted how hydrophobic residues tend to

mutate to other hydrophobic ones, but is this the case also for the subset of resistant mutations? I wonder if the data might reveal some interesting patterns about what kind of mutations (based on the physicochemical properties of the residues involved, as well as their location with respect to the ligand) are generally associated with resistance. This kind of analysis could also be performed as part of a feature importance analysis for some of the ML models trained. In addition, during the discussion of amino acid substitutions at pages 5 and 6, it would be beneficial to discuss what kind of factors affect these distributions, and in particular what biases might be present in the dataset.

In summary, I think MdrDB is a useful contribution to the chem/bio-informatics community, but a more comprehensive analysis of the data and ML models would enhance the profile of the work.

Reviewer #2 (Remarks to the Author):

The authors developed a new database MdrDB that integrates data from seven publicly available datasets to provide a comprehensive resource for studying mutation-induced drug resistance. It includes information on drug sensitivity and cell line mutations and comprises 100,537 samples of 240 proteins, 2,503 mutations, and 440 drugs. MdrDB provides 3D structures of wild type and mutant protein-ligand complexes, binding affinity changes upon mutation, and biochemical features. The database has been shown to enhance the performance of commonly used machine learning models when predicting binding affinity changes. Overall, MdrDB is a valuable tool for advancing the understanding of mutation-induced drug resistance and accelerating drug discovery. However, the manuscript does have a few issues that need to be addressed:

- In the Data Collection section, the data sources were briefly described, but the specific details of the provided data were not indicated. For each data source, it was not clear whether the complex structures of the wild-type or mutant were provided, as well as the types of mutations present in each dataset.
- The statement “Mutations were then filtered to retain only six types of mutation: single substitution, multiple substitution, deletion, insertion, indel and complex mutations (i.e., combinations of the other five mutation types)” is unclear. The term "single substitution" may refer to a type of mutation where a single nucleotide base is replaced with a different nucleotide base in DNA, or a single amino acid is replaced with a different amino acid in a protein. The same applies to "multiple substitution". Furthermore, what types of variants did the authors investigate at the protein level? For example, single nucleotide substitutions can lead to missense, nonsense, and silent mutations. Therefore, the authors should provide a detailed and accurate description of the types of mutations studied.

- Please carefully check the accuracy of the $\Delta\Delta G$ formula. The formula for binding affinity calculation is $\Delta G_{exp} = RT \ln(K_i/IC_{50})$
- The statement “After this, for each (protein, cell line, drug) mutant sample, the mutation string was generated by merging all mutations of the mutant protein; IC50 values were averaged over all cell lines, and this average value was taken as the mutant protein-drug affinity value IC50(MT). A corresponding wild type sample was similarly assigned and IC50s again averaged over cell lines, and this average value taken as the wild type protein-drug affinity IC50(WT).” raises two questions: (1) Is it reasonable to apply the mean value, and how are outliers handled? (2) For different mutation samples, the types of mutations may be different. In such cases, was the calculation of the mean value performed based on (protein, cell line, drug and mutations)?
- The statement “For samples from Platinum and TKI, wild type and mutant protein-drug complex structures were also obtained from the original datasets.” is unclear. Do Platinum and TKI also include all mutant protein-drug complex structures? Please verify.
- The statement “Meanwhile, the largest ligand was kept as the docking box generation reference.” is ambiguous. Is the largest ligand the one that will be studied? If not, how can you determine that they have the same binding pocket? In some cases, a crystal structure may contain multiple ligands that occupy different binding sites on the protein.
- The established model should be compared to other reported methods, such as PremPLI.
- Please provide a definition for the term 'sample' in this context.

Reviewer #3 (Remarks to the Author):

In this study, the authors have developed a drug resistance database named MdrDB, which is highly associated with protein structure mutations. The authors have gathered protein-drug binding affinity data from seven public databases, and have also collected the protein-drug complex structure from PDB database. In cases where the complex structure was unavailable, they have expertly modeled it by AF2 protein structure prediction and molecular docking techniques. MdrDB database is a valuable resource that provides a wealth of information about various protein and drug features. Significantly, the authors have used the MdrDB core dataset to train their machine learning model, which has demonstrated superior performance when compared to other methods based on different datasets. This remarkable result indicates the essential role played by MdrDB core dataset in enhancing the accuracy of the model

predictions. Overall, this paper is a well-written and systematic work that presents a valuable contribution to the field of drug resistance research. I suggest that this paper can be accepted after solving the following issues:

1. The authors claim that “Protein-ligand binding affinity data is of great value for understanding the impact of polymorphisms on disease and identifying mutations that lead to drug resistance”, however, some literature reference would be useful to add for those interested in.
2. Association / disassociation / equilibrium / inhibition constants are also commonly used parameters to describe the drug-target binding affinity. Did MdrDB collect only the reported cases with IC50 values? The IC values should be provided in each sample in MdrDB.
3. The authors validated the MdrDB by construct the machine learning models, however, more details on the validation protocol should be added.
4. For a protein with multiple structures in RCSB PDB database, how did MdrDB deal with these structures?
5. MdrDB does not support search by keyword of protein name. Search by protein name would be helpful for users to find their interest drug targets.
6. Searching by drug structure may be convenient for users to find the related information.
7. I could not find the drug binding poses in MdrDB, such as MdrDB000001, MdrDB000002, MdrDB000003. It is better to show the 3D structure of protein-drug complex.
8. In MdrDB sample details, the unit should be marked in the items, such as DDG, Volume, MolWt
9. In MdrDB download webpage, it is better to allow uses to download the drug structures (docking poses) files.
10. A related reference for protein mutation prediction should be cited, J. Phys. Chem. B 2010, 114, 9663-9676

May 12, 2023

Re: Paper # COMMSCHEM-23-0138-A – “MdrDB: Mutation-induced drug resistance DataBase” by Yang et al.

Thank you for your help in handling our paper. We are grateful for the reviewers' constructive comments, which have been helpful in improving or clarifying aspects of our manuscript, and we have made earnest efforts to address them. We hope that the revised manuscript now meets the high standards set by Communications Chemistry. We appreciate your further consideration of our manuscript.

Yours sincerely,

Ziyi Yang

Senior Researcher

Tencent Quantum Laboratory,

Shenzhen 518057, Guangdong, China

[E-mail: chriszyyang@tencent.com](mailto:chriszyyang@tencent.com)

In our responses to the reviewers' comments, we have color-coded the reviewer's questions in **black**, our responses in **blue**, and the revised manuscript amendments in **red**.

Reviewer #1 (Remarks to the Author):

Yang et al. present a new database (MdrDB) containing protein-ligand binding affinity changes for a set of ~2500 mutations across >200 proteins. MdrDB was created by consolidating data from 7 datasets with a custom data processing pipeline. Structural information was included using experimental data from the Protein Data Bank or AlphaFold2 predictions. Pre-computed features that can be used as input for machine learning models are also provided, which is useful to create a standardized benchmark for descriptor-based models. The significance of this effort lies in the creation of a consistent and machine-readable dataset of mutational affinity values that is considerably larger than existing datasets, and one that includes multiple and complex (e.g., insertions and deletions) mutations. Overall, the manuscript is clear, concise, and well written.

For the above reasons, my opinion is that this work is in principle publishable in Communications Chemistry. At the same time, I felt that the analysis of the dataset and, in particular, the exploration of ML models was fairly superficial and with room for improvement. My two main suggestions are the following:

Response: We appreciate that the reviewer agrees this manuscript could be a meaningful contribution to the community.

1. While MdrDB contains over 200 protein targets and ~2500 mutations, only a small subset (the TKI set with ~140 mutations) was used to evaluate ML models. While the TKI work is interesting to show how more data results in a better model, this seems like a missed opportunity to utilize the newly created dataset to evaluate the ability of ML to predict drug resistance more broadly across protein classes and mutation types (single, multiple, complex).

Many additional interesting insights may arise from such a benchmark, e.g., is there a difference in predictive performance between mutations with experimental vs predicted structures? Which kind of mutations are more challenging to predict with ML?

How far from the training set are these models able to extrapolate? One can imagine splitting the dataset in a few different ways to test ability to extrapolate in protein, ligand, or mutation space. These are only a few examples of possible questions one could explore with MdrDB. In addition, creating multiple train/test splits (like in cross validation) would allow to compute uncertainties for the performance metrics used (i.e., adding error bars to the plots in Figure 3A).

I understand that a comprehensive study may be beyond the scope of this work, but I think it would still be worth exploring a couple of hypotheses while making use of the full dataset.

Response: Thank you for your comments and suggestions. In the original manuscript, we conducted experiments on the TKI dataset to investigate whether utilizing MdrDB_Coreset as the training dataset can enhance model generalization performance for predicting mutation-induced drug resistance in the TKI dataset, which could be compared with previous studies. However, we agree with the reviewer, that utilizing the constructed MdrDB database to provide a more comprehensive evaluation of machine learning methods and provide benchmark prediction results would help to extend the impact of our work, and enable more researchers to use MdrDB for developing novel machine learning algorithms and facilitating drug resistance research.

We have now further investigated the baseline results of 10 commonly used machine learning methods on the MdrDB_CoreSet, in four scenarios:

Scenario 1 benchmarks the performances of different ML methods on the single substitution subset of MdrDB when using different train-test splitting settings. The samples can be randomly split (1.1), 5-fold cross-validated (1.2), split with protein information (1.3, 1.4), split with drug information (1.5, 1.6), and split with mutation information (1.7, 1.8). Scenario 2 benchmarks the performances of ML methods on multiple substitution subset of MdrDB using randomly split (2.1), and 5-fold cross-validation (2.2).

In the final two scenarios we evaluate the extrapolation ability of the ML models trained on single substitution mutations to complex mutations. In scenario 3, we directly evaluate the performances on the multiple substitution subset (3.1) and the remaining mutation types (3.2). In scenario 4, we first used 80% of the samples in other mutation subsets to fine-tune the ML models and used the remaining 20% for evaluation.

Overall, we find that tree-based methods (in particular, ExtraTrees) typically obtain better prediction performance than other types of methods. However, current ML methods still have major limitations on the task of inferring from single substitution mutations to predicting drug resistance caused by other mutations.

For further details, please see the model performance evaluation section of the Supplementary Material. In addition, we provided several Jupyter notebooks for these scenarios, which are available at <https://github.com/tencent-quantum-lab/MdrDB>. In addition to computing uncertainties for the performance metrics, we report the mean and standard deviation for each scenario. The corresponding code is available at <https://github.com/tencent-quantum-lab/MdrDB>.

Action: In Page 8, in the model performance evaluation section of the revised manuscript, we made the following change:

Add “In addition, we also provide a comprehensive evaluation of 10 machine learning models in several different scenarios, and provide baseline prediction results on the MdrDB database. It is our hope that this will help further the development of new machine learning algorithms using the MdrDB database and facilitate drug resistance research. Please refer to the Supplementary Material for details.”

At the suggestion of the reviewer, we added error bars in Figure 3A.

2. It would be useful to expand the analysis of the dataset to discuss the statistics of mutations for the class of resistant mutations only and how that differs from the distribution of non-resistant mutations. For example, at page 7 it was noted how hydrophobic residues tend to mutate to other hydrophobic ones, but is this the case also

for the subset of resistant mutations? I wonder if the data might reveal some interesting patterns about what kind of mutations (based on the physicochemical properties of the residues involved, as well as their location with respect to the ligand) are generally associated with resistance. This kind of analysis could also be performed as part of a feature importance analysis for some of the ML models trained.

In addition, during the discussion of amino acid substitutions at pages 5 and 6, it would be beneficial to discuss what kind of factors affect these distributions, and in particular what biases might be present in the dataset.

Response: For amino acid substitutions, we compared the frequency of mutations for each amino acid type with the expected frequency calculated using codon frequencies from a previous study [1]. We found a high correlation between the amino acid frequency at the mutation site and the expected frequency (Pearson = 0.945), implying that the mutations were essentially random. On the other hand, the frequency of amino acids after mutations had only a moderate correlation with the expected frequency (Pearson = 0.633). Many factors may contribute to this besides codon frequencies, such as diseases, epigenetics, and experimental assays.

In addition, based on the reviewer's suggestion, we also compared the frequency of mutations for each amino acid type in the resistant/susceptible samples with codon frequency calculated by study [1]. We found a high correlation (Pearson = 0.944) between the amino acid frequency before mutation in susceptible samples of MdrDB and the expected codon frequency. For the resistant samples, the amino acid frequency before mutation has a relatively strong correlation with the expected codon frequency (Pearson = 0.874), where serine (S), lysine (K) and tyrosine (Y) varies a lot. Mutated amino acid frequencies were moderately correlated with expected codon frequencies in both susceptible (Pearson = 0.617) and resistant samples (Pearson = 0.65), however, the distributions of frequencies are quite different.

Action: In Page 7, database summary section of the revised manuscript, we made the following change:

Add “The mutation frequencies of different amino acids at the mutation sites are highly correlated with the frequencies calculated based on codon frequencies (Pearson $r = 0.945$, Figure S3A and Table S6)⁴⁰. However, the frequencies of amino acids after the mutation showed a lower correlation to the frequencies calculated based on codon frequencies (Pearson $r = 0.633$, Figure S3B)⁴⁰. More complex factors could influence this, such as epigenetic modifications⁴¹, disease preferences⁴², co-evolution⁴³, and experimental assay biases⁴⁴. In addition, we analyzed the mutation spectrums for each type of amino acid, which show the mutation preferences. Interestingly, as shown in Figure S4, most of the amino acids have a mutation spectrum that is similar to the one from the codon frequency calculation (Pearson $r > 0.8$). For the deviated ones, the higher ratio of lysine (K), glutamine (Q), cysteine (C), and tryptophan (W) are mutated to A, which may come from the loss of function mutation experiments, like alanine scanning mutagenesis⁴⁴. We further carried out the same analyses on susceptible/resistant samples (Figure S5). Interestingly, we could observe an obvious decrease in the frequency correlation on resistance samples before mutation (Pearson $r = 0.874$), where serine (S), lysine (K), and tyrosine (Y) showed the largest differences. For the frequency of each amino acid after mutation in the susceptible/resistance samples, although the correlations are similar, their distribution varies greatly.”

Corresponding figures were added to the Supplementary Material.

Figure S3. Statistical analysis between the frequency of each amino acid type before and after mutation in MdrDB and the expected frequency calculated using codon frequencies from a previous study. A (left column) is before mutation, B (right column) is after mutation. (Top) The expected codon frequency of each amino acid; (Middle) The frequency of each amino acid type in MdrDB; (Bottom) Correlation between the frequency of each amino acid type before mutation in MdrDB and the expected frequency.

Figure S4. Correlation between the mutation spectrum of each amino acid in the MdrDB single substitution subset and the mutation spectrum calculated based on codon frequency.

Figure S5. Correlation between the frequency of each amino acid before and after mutation in the susceptible (left column) /resistant (right column) samples with the expected codon frequency. (A) Wild type (i.e., before mutation); (B) Mutation; (C) The wild-type frequency of each amino acid type in susceptible and resistant samples of MdrDB; (D) The mutation frequency of each amino acid type in susceptible and resistant samples of MdrDB.

In summary, I think MdrDB is a useful contribution to the chem/bio-informatics community, but a more comprehensive analysis of the data and ML models would enhance the profile of the work.

Response: We appreciate that the reviewer agrees this manuscript could be a meaningful contribution to the community. We thank the reviewer again for the helpful comments and suggestions, which we have made earnest efforts to address.

Reviewer #2 (Remarks to the Author):

The authors developed a new database MdrDB that integrates data from seven publicly available datasets to provide a comprehensive resource for studying mutation-induced drug resistance. It includes information on drug sensitivity and cell line mutations and comprises 100,537 samples of 240 proteins, 2,503 mutations, and 440 drugs. MdrDB provides 3D structures of wild type and mutant protein-ligand complexes, binding affinity changes upon mutation, and biochemical features. The database has been shown to enhance the performance of commonly used machine learning models when predicting binding affinity changes. Overall, MdrDB is a valuable tool for advancing the understanding of mutation-induced drug resistance and accelerating drug discovery. However, the manuscript does have a few issues that need to be addressed:

Response: We appreciate that the reviewer agrees this manuscript could be a meaningful contribution to the community.

1. In the Data Collection section, the data sources were briefly described, but the specific details of the provided data were not indicated. For each data source, it was not clear whether the complex structures of the wild-type or mutant were provided, as well as the types of mutations present in each dataset.

Response: Thank you for your comments and suggestions. We have added the specific details of the data sources in MdrDB in Table S1. In addition, we provided the source information of each sample on the sample details page of the MdrDB database website, including data sets, mutations, and drug poses, to facilitate researchers to trace the data.

Action: In Page 7, the database summary section of the revised manuscript, we have made the following changes.

Add "Table 1 summarizes the data contained in MdrDB, and an overview of the data sources is shown in Table S1 in Supplementary Material."

The supplementary table is attached in the Supplementary Material.

Table S1. Overview of data sources in the MdrDB.

Data source		AIMMS	DepMap	GDSC	KinaseMD	Platinum	RET	TKI
Mutant structure source	# Processed	0	0	0	0	840	0	144
	# PyMOL mutated	5048	1347	86802	1334	0	456	0
	# AlphaFold2 folded	0	153	4413	0	0	0	0
Drug pose source	# Co-crystal	0	0	0	0	840	0	131
	# Docked	5048	1500	91215	1334	0	456	13
Mutation types	# Single substitution	224	1343	80190	992	689	456	144
	# Multiple substitution	4824	4	6656	342	151	0	0
	# Deletion	0	153	3075	0	0	0	0
	# Insertion	0	0	355	0	0	0	0
	# Indel	0	0	243	0	0	0	0
	# Complex	0	0	696	0	0	0	0

2. The statement “Mutations were then filtered to retain only six types of mutation: single substitution, multiple substitution, deletion, insertion, indel and complex mutations (i.e., combinations of the other five mutation types)” is unclear. The term "single substitution" may refer to a type of mutation where a single nucleotide base is replaced with a different nucleotide base in DNA, or a single amino acid is replaced with a different amino acid in a protein. The same applies to "multiple substitution". Furthermore, what types of variants did the authors investigate at the protein level? For example, single nucleotide substitutions can lead to missense, nonsense, and silent mutations. Therefore, the authors should provide a detailed and accurate description of the types of mutations studied.

Response: We thank the reviewer for pointing out an ambiguous description of the mutation types. In MdrDB, we only considered mutations of amino acids in the proteins.

Action: We have clarified this point in the mutation descriptions on Page 3, the data preprocessing section of the revised manuscript, when the mutations are first mentioned.

“Mutations were then filtered to retain only six types of mutation: single substitution, multiple substitution, deletion, insertion, indel and complex mutations (i.e., combinations of the other five mutation types)” is changed to “Mutations were then filtered to retain only six types of mutation in terms of amino acid changes in protein sequences:

- **Single substitution mutation: replacement of one amino acid in a protein with a different amino acid.**
- **Multiple substitution mutation: more than one amino acid substitution in a protein.**
- **Deletion mutation: removal of one amino acid or an amino acid sequence in a protein.**
- **Insertion mutation: addition of one amino acid or an amino acid sequence in a protein.**
- **Indel mutation: a deletion mutation followed by an insertion mutation, i.e., replacement of an amino acid sequence in a protein with another amino acid sequence.**
- **Complex mutation: combinations of the above five amino acid mutation types.”**

3. Please carefully check the accuracy of the $\Delta\Delta G$ formula. The formula for binding affinity calculation is $\Delta G_{\text{exp}} = RT \ln(K_i/IC_{50})$.

Response: We thank the reviewer for pointing out this error. We had cited an incorrect paper and used the wrong formula.

Action: We have corrected the reference and formula on Page 4, the data preprocessing section of the revised manuscript.

Finally, these affinity pairs were used to calculate $\Delta\Delta G$ according to the formula^{18, 29}:

$$\Delta\Delta G_{\text{exp}} = RT \ln \frac{K_{i,\text{mut}}}{K_{i,\text{WT}}} \approx RT \ln \frac{IC_{50,\text{mut}}}{IC_{50,\text{WT}}}$$

4. The statement “After this, for each (protein, cell line, drug) mutant sample, the mutation string was generated by merging all mutations of the mutant protein; IC50 values were averaged over all cell lines, and this average value was taken as the mutant protein-drug affinity value IC50(MT). A corresponding wild type sample was similarly assigned and IC50s again averaged over cell lines, and this average value taken as the wild type protein-drug affinity IC50(WT).” raises two questions: (1) Is it reasonable to

apply the mean value, and how are outliers handled? (2) For different mutation samples, the types of mutations may be different. In such cases, was the calculation of the mean value performed based on (protein, cell line, drug and mutations)?

Response: For the first question, we followed the data processing protocol of the KinaseMD database, using the mean values of IC50s from different drug-treated cell lines. In addition, we calculated the mean number of wild-type cell lines used to calculate IC50(WT) and the standard deviation of these values as 200 and 3×10^{-16} , respectively. Therefore, we consider it reasonable to use the mean value for IC50(WT), to which we do not apply additional outlier handling. However, we found that there was just one cell line for the mutant in most of the samples to be used as IC50(MT), which also made the outlier analyses infeasible. Although we used a simple processing method to calculate the IC50, we thought that users might be more interested in processing the data themselves. Therefore, in order to make the data traceable, we put the details and data of the cell lines used for the calculation in each sample on the download page of the MdrDB website.

For the second question, we apologize for the confusion in the description. Mutation strings are generated by checking the mutation information of proteins in a specific cell line. Therefore, the mean value is calculated based on averaging cell lines of (protein, cell line, drug, mutations) samples. We have modified the corresponding description on Page 3, the data preprocessing section of the revised manuscript, to make this clear.

Action: We have modified the corresponding descriptions on Page 3, the data preprocessing section of the revised manuscript to make things clear.

“After this, for each (protein, cell line, drug) mutant sample, the mutation string was generated by merging all mutations of the mutant protein; IC50 values were averaged over all cell lines, and this average value was taken as the mutant protein-drug affinity value IC50(MT).” is changed to “After this, for each (protein, cell line, drug) mutant sample, the mutation string was generated by merging all mutations of the mutant protein to get (protein, cell line, drug, mutation) sample. Then, IC50 values were

averaged over all cell lines for the (protein, cell line, drug, mutation) sample following the sensitivity data processing in KinaseMD²². This average value was taken as the mutant protein-drug affinity value IC₅₀ (MT).”

5. The statement “For samples from Platinum and TKI, wild type and mutant protein-drug complex structures were also obtained from the original datasets.” is unclear. Do Platinum and TKI also include all mutant protein-drug complex structures? Please verify.

Response: Yes, the Platinum and TKI databases provide structural information for wild-type and mutant protein-ligand complexes. Therefore, we directly incorporate these structural data into our database.

6. The statement “Meanwhile, the largest ligand was kept as the docking box generation reference.” is ambiguous. Is the largest ligand the one that will be studied? If not, how can you determine that they have the same binding pocket? In some cases, a crystal structure may contain multiple ligands that occupy different binding sites on the protein.

Response: In this step, the ligands associated with a specific chain are extracted. As shown in Figure 1, in most cases only one ligand (38.7 %) or no ligand (49.7 %) is associated with a protein chain. For the remaining cases, we kept the largest ligand to define the docking box in the current version of MdrDB. As the reviewer mentioned, in some cases, there may be multiple ligands bound to the same chain. For these cases, the largest ligand would not work, and a docking box defined directly with multiple ligands is more suitable. We appreciate your insightful suggestions and will update this in newer versions of MdrDB.

Peer Review Figure 1. Histogram of the number of ligands in a single protein chain in MdrDB.

7. The established model should be compared to other reported methods, such as PremPLI.

Response: Following reference [1], we found that PremPLI uses a random forest regression scoring function and consists of 11 sequence-based and structure-based features. The baseline methods reported in our manuscript also include random forest regression.

In MdrDB, based on the structures of the wild type and mutant protein-ligand complexes, we calculated a total of 146 biochemical features relevant to machine learning prediction of mutation-induced affinity changes, which will help more researchers develop new machine learning algorithms using the MdrDB database and facilitate resistance research. In addition, if researchers are interested in this problem, we provide the 3D structure of wild-type and mutant protein-ligand complexes that can be used to calculate features for machine learning to predict drug resistance.

Reference:

[1] Sun, T., Chen, Y., Wen, Y., Zhu, Z., & Li, M. (2021). PremPLI: a machine learning model for predicting the effects of missense mutations on protein-ligand interactions. *Communications biology*, 4(1), 1311.

8. Please provide a definition for the term 'sample' in this context.

Response: For the MdrDB_Fullset, a sample is defined in terms of “Uniprot-PDB-mutation-drug”. For the MdrDB_CoreSet, Non-repetitive “Uniprot-mutation-drug” is

defined as a sample. These definitions are given in the download section of the MdrDB tutorial, and we have now added this information in the revised manuscript.

Action: In Page 6, Figure 2 of the revised manuscript, we made the following change. Add “where a sample corresponds to a (Uniprot, PDB, mutation, drug) combination”. In Figure S2 of Supplementary material, we added “Non-repetitive “Uniprot-mutation-drug” is defined as a sample”.

Reviewer #3 (Remarks to the Author):

In this study, the authors have developed a drug resistance database named MdrDB, which is highly associated with protein structure mutations. The authors have gathered protein-drug binding affinity data from seven public databases, and have also collected the protein-drug complex structure from PDB database. In cases where the complex structure was unavailable, they have expertly modeled it by AF2 protein structure prediction and molecular docking techniques. MdrDB database is a valuable resource that provides a wealth of information about various protein and drug features. Significantly, the authors have used the MdrDB core dataset to train their machine learning model, which has demonstrated superior performance when compared to other methods based on different datasets. This remarkable result indicates the essential role played by MdrDB core dataset in enhancing the accuracy of the model predictions. Overall, this paper is a well-written and systematic work that presents a valuable contribution to the field of drug resistance research. I suggest that this paper can be accepted after solving the following issues:

Response: We appreciate that the reviewer agrees this manuscript could be a meaningful contribution to the community.

1. The authors claim that “Protein-ligand binding affinity data is of great value for understanding the impact of polymorphisms on disease and identifying mutations that

lead to drug resistance”, however, some literature references would be useful to add for those interested in.

Response: Thank you for your comments and suggestions. We have added some references in the revised manuscript.

Action: In Page 1, the Introduction section of the revised manuscript, we made the following change.

Add references: “Protein-ligand binding affinity data is of great value for understanding the impact of polymorphisms on disease and identifying mutations that lead to drug resistance [18,19].”

2. Association / disassociation / equilibrium / inhibition constants are also commonly used parameters to describe the drug-target binding affinity. Did MdrDB collect only the reported cases with IC50 values? The IC values should be provided in each sample in MdrDB.

Response: We checked the binding affinity of the raw data. In the collected data sets, some recorded binding affinity of the wild type and the mutant type were measured experimentally by IC50 (e.g., GDSC); some datasets directly provide experimental binding affinity change values DDG (e.g., AIMMS); some datasets provide the binding affinity change value fold-change, which can be converted to DDG (e.g., Platinum). Therefore, the MdrDB database uses DDG as a unified standard to calculate the wild-type and mutant binding affinity change values and does not provide the wild-type and mutant affinity values corresponding to each sample, respectively.

3. The authors validated the MdrDB by construct the machine learning models, however, more details on the validation protocol should be added.

Response: In the revised manuscript, we further provide a comprehensive evaluation of 10 common machine learning methods in several scenarios and provide baseline prediction results on the MdrDB database, which will help more researchers develop

new machine learning algorithms using the MdrDB database and facilitate resistance research.

Briefly, we first added a section (Scenario 1) to benchmark the performances of different ML methods on the single substitution subset of MdrDB when using different train-test splitting settings. The samples can be randomly split (1.1), 5-fold cross-validated (1.2), split with protein information (1.3, 1.4), split with drug information (1.5, 1.6), and split with mutation information (1.7, 1.8).

Similarly, we added Scenario 2 to benchmark performances of ML methods on multiple substitution subset of MdrDB using randomly split (2.1), and 5-fold cross-validation (2.2).

Then, we evaluate the extrapolation ability of the ML models trained on single substitution to complex mutation. In scenario 3, we directly evaluate the performances on the multiple substitution subset (3.1) and the rest mutation type subsets (3.2). In scenario 4, we first used 80% of the samples in other mutation subsets to fine-tune the ML models and used the rest 20% for evaluation.

Overall, tree-based methods typically could obtain better prediction performance than other types of methods, especially ExtraTrees. However, current ML methods still have major limitations on the task of inferring from single substitution mutations to predicting drug resistance caused by other mutations.

Thanks again for the review's comments. In the revised manuscript, we further investigated the baseline results of MdrDB in four scenarios. For the details, please see the model performance evaluation section in Supplementary Material, and we provided several Jupyter notebooks for these scenarios, which are available at <https://github.com/tencent-quantum-lab/MdrDB>.

Action: In Page 8, model performance evaluation section of the revised manuscript, we made the following change:

Add "In addition, we also provide a comprehensive evaluation of machine learning methods through several scenarios and provide baseline prediction results on the MdrDB database, which will help more researchers develop new machine

learning algorithms using the MdrDB database and facilitate resistance research. To save space, please refer to Supplementary Material for details.”

In the section of model performance evaluation of Supplementary Material, we reported the baseline results of 10 commonly used machine learning methods on the MdrDB_Coreset in above mentioned varying experimental scenarios, and the corresponding code is available at <https://github.com/tencent-quantum-lab/MdrDB>.

4. For a protein with multiple structures in RCSB PDB database, how did MdrDB deal with these structures?

Response: For each Uniprot ID, all associated PDBs were identified with the RCSB REST API. The .pdb or .cif for 3D structures and .fasta for sequences were downloaded from RCSB PDB. After that, we preprocessed the structure files, for protein PDB files, all water, solvent, and ions were removed. Then, the protein chains and ligands were split into separate files. Each chain was annotated and only the chains corresponding to the protein were kept. If multiple chains exist for the protein, the longest one was kept. Each mutation for a protein was checked against all available PDBs. If the mutation sites could be found in the PDB, the mutation and drug would be assigned to the PDB. Therefore, the "Uniprot-mutation-drug" sample contains multiple data generated based on different PDB structures. The MdrDB tutorial (<https://quantum.tencent.com/mdrdb/tutorial/>) shows the details of structure file preprocessing.

5. MdrDB does not support search by keyword of protein name. Search by protein name would be helpful for users to find their interested drug targets.

Response: We have now added the ability to search by keyword of protein name to the MdrDB website. For example, users can type “EGFR” into the search bar, and the corresponding samples can be obtained directly.

6. Searching by drug structure may be convenient for users to find the related information.

Response: On the MdrDB website, we provide several types of customized search, such as by SMILES string, mutation type, PDB ID, and the source database. The general format of a query using the advanced keywords takes the form: `prefix: search_content`. If the user wants to search by SMILES string, they can enter S:*COC1* in the search term. The details of advanced search are shown in the tutorial of MdrDB (<https://quantum.tencent.com/mdrdb/tutorial/>).

0. I could not find the drug binding poses in MdrDB, such as MdrDB000001, MdrDB000002, MdrDB000003. It is better to show the 3D structure of protein-drug complex.

Response: On the sample detail page of the MdrDB website we now provide 3D structures of the wild-type and mutant protein-ligand complexes.

1. In MdrDB sample details, the unit should be marked in the items, such as DDG, Volume, MolWt.....

Response: In the sample detail page of the MdrDB website, we have we have added unit information to some items, such as DDG, and MolWt.

2. In MdrDB download webpage, it is better to allow users to download the drug structures (docking poses) files.

Response: On the download page of the MdrDB website, we provide the structure file of MdrDB for users to download. Processed wild-type and mutant protein structures are grouped by mutation types. For each type, the corresponding samples and overall feature data (.tsv) are included in the .tar.gz file. Each sample contains 5 structure files (i.e., wild-type protein, mutant protein, drug, wild-type protein-drug complex, and mutant protein-drug complex).

3. A related reference for protein mutation prediction should be cited, J. Phys. Chem. B 2010, 114, 9663-9676.

Response: We have added this reference to Page 8, the model performance evaluation section of the revised manuscript.

REVIEWERS' COMMENTS:

Reviewer #1 (Remarks to the Author):

The authors addressed the concerns I had raised and open-sourced the code related to these additional analyses. Therefore, my opinion is that the manuscript is now ready for publication.

Reviewer #2 (Remarks to the Author):

The author has addressed most of my questions except for the one below. Platinum only offers a limited selection of mutant protein-drug complex structures. However, I was unable to locate any mutant protein-drug complex structures for TKI. Could you kindly provide the source for such structures?

5. The statement "For samples from Platinum and TKI, wild type and mutant protein- drug complex structures were also obtained from the original datasets." is unclear. Do Platinum and TKI also include all mutant protein-drug complex structures? Please verify.

Response:

Yes, the Platinum and TKI databases provide structural information for wild-type and mutant protein-ligand complexes. Therefore, we directly incorporate these structural data into our database.

Reviewer #3 (Remarks to the Author):

The authors have solved my issues. I think this paper could be considered to be published in this stage.

May 29, 2023

Re: Paper # COMMSCHEM-23-0138B – “A mutation-induced drug resistance database (MdrDB)” by Yang et al.

Thank you for your help in handling our paper. We thank reviewer #1 and reviewer #3 for agreeing that our manuscript can be published at this stage, and we have carefully addressed the remaining issues of reviewer #2. We hope that the revised manuscript now meets the high standards set by Communications Chemistry. Furthermore, we appreciate your agreeing to publish a suitably revised version in Communications Chemistry.

Yours sincerely,

Ziyi Yang

Senior Researcher

Tencent Quantum Laboratory,

Shenzhen 518057, Guangdong, China

[E-mail: chriszyyang@tencent.com](mailto:chriszyyang@tencent.com)

In our responses to the reviewers' comments, we have color-coded the reviewer's questions in **black**, and our responses in **blue**.

Reviewer #1 (Remarks to the Author):

The authors addressed the concerns I had raised and open-sourced the code related to these additional analyses. Therefore, my opinion is that the manuscript is now ready for publication.

Response: We appreciate that the reviewer agrees that this manuscript is acceptable for Communications Chemistry.

Reviewer #2 (Remarks to the Author):

1. The author has addressed most of my questions except for the one below. Platinum only offers a limited selection of mutant protein-drug complex structures. However, I was unable to locate any mutant protein-drug complex structures for TKI. Could you kindly provide the source for such structures?

Response: The wild-type and mutant protein-ligand complex structures of the TKI dataset are attached to the supplementary material of Aldeghi et al [1]. In addition, our database also provides structure files of TKI datasets. Users can search for samples belonging to the TKI dataset in the data source column on the search page of MdrDB, and download related structure files on the sample detail page.

Reference:

[1] Aldeghi, Matteo, Vytautas Gapsys, and Bert L. de Groot. "Predicting kinase inhibitor resistance: physics-based and data-driven approaches." *ACS central science* 5.8 (2019): 1468-1474.

2. The statement "For samples from Platinum and TKI, wild type and mutant protein-drug complex structures were also obtained from the original datasets." is unclear. Do Platinum and TKI also include all mutant protein-drug complex structures? Please verify.

Response: Yes, the Platinum and TKI databases provide structural information for wild-type and mutant protein-ligand complexes. In MdrDB, we incorporate these

structural data directly into our database.

Reviewer #3 (Remarks to the Author):

The authors have solved my issues. I think this paper could be considered to be published in this stage.

Response: We appreciate that the reviewer agrees that this manuscript is acceptable for Communications Chemistry.